# Communication-Efficient Tracking of Unknown, Spatially Correlated Signals in Ad-Hoc Wireless Sensor Networks: Two Machine Learning Approaches

**DOI:** 10.3390/s21155175

**Published:** 2021-07-30

**Authors:** Hadi Alasti

**Affiliations:** School of Polytechnic, College of Engineering, Purdue University Fort Wayne, Fort Wayne, IN 46805, USA; halasti@Purdue.edu or halasti@ieee.edu

**Keywords:** machine learning, spatial signal modeling, spatial tracking, signal processing, ad-hoc sensor network

## Abstract

A low-cost machine learning (ML) algorithm is proposed and discussed for spatial tracking of unknown, correlated signals in localized, ad-hoc wireless sensor networks. Each sensor is modeled as one neuron and a selected subset of these neurons are called to identify the spatial signal. The algorithm is implemented in two phases of spatial modeling and spatial tracking. The spatial signal is modeled using its *M* iso-contour lines at levels ${\left\{\ell_{j}\right\}}_{j=1}^{M}$ and those sensors that their sensor observations are in Δ margin of any of these levels report their sensor observations to the fusion center (FC) for spatial signal reconstruction. In spatial modeling phase, the number of these contour lines, their levels and a proper Δ are identified. In this phase, the algorithm may either use adaptive-weight stochastic gradient or scaled stochastic gradient method to select a proper Δ. Additive white Gaussian noise (AWGN) with zero mean is assumed along with the sensor observations. To reduce the observation noise’s effect, each sensor applies moving average filter on its observation to drastically reduce the effect of noise. The modeling performance, the cost and the convergence of the algorithm are discussed based on extensive computer simulations and reasoning. The algorithm is proposed for climate and environmental monitoring. In this paper, the percentage of wireless sensors that initiate a communication attempt is assumed as cost. The performance evaluation results show that the proposed spatial tracking approach is low-cost and can model the spatial signal over time with the same performance as that of spatial modeling.

## 1. Introduction

This paper presents a machine learning (ML) algorithm for recognition and low-cost tracking of unknown spatially correlated signals using sensor readings in ad-hoc wireless sensor fields. The randomly distributed wireless sensors are modeled as neurons and subsets of these neurons are selected to identify the unknown signal. In this identification problem, the signal is modeled using its *M* iso-contour lines at levels ${\left\{\ell_{j}\right\}}_{j=1}^{M}$. Modeling the spatial signal using their contour levels has been used in several applications, such as medical imaging [1,2]; geographic information systems [3]; computer vision [4]; etc. In wireless sensor network, modeling the spatial signals using their contour lines compresses the signal to a limited number of sensor readings, where as result it conserves massive amount of in-network energy and can increase the network’s lifetime. Energy conservation is a challenging problem in wireless sensor networks [5].

The proposed algorithm has applications in environmental monitoring, such as monitoring the temperature of heat-island [6], gas density monitoring [7,8], monitoring the city air pollution [9,10,11,12], smart agriculture [13,14], smart battlefield [15]; where the objective is to monitor the distribution of a correlated physical quantity such as density of gasses, pollutants, radiations, moisture, temperature, etc. In modern days the smart Internet of things (IoT) devices can act as nodes of sensor network for monitoring of the desired quantities in the extent of a vast area, such as extent of a city, a forest, or even a deserted area. Study of the spatiotemporal distribution of the number of infections to a contagious disease such as COVID-19 [16,17,18] within a large area of a country is another application example of the discussed algorithm in this paper.

In this paper, a cost-efficient algorithm is proposed and discussed for spatial monitoring of unknown, correlated signals over time from wireless sensor observations. Localization of the sensor nodes and the correlation in spatial signal are the only assumptions from the sensor field. Two machine learning (ML) algorithms based on stochastic gradient are used to derive the spatial model parameters. The spatial signal is modeled using its *M* contour lines at levels ${\left\{\ell_{j}\right\}}_{j=1}^{M}$ and those wireless sensors that their sensor observations are within Δ margin of any of these contour levels report their sensor observations to the fusion center(s) (FC) for spatial signal reconstruction. The proposed algorithm is implemented in spatial modeling, and spatial tracking phases. In spatial modeling phase, the model parameters, Δ and ${\left\{\ell_{j}\right\}}_{j=1}^{M}$ are identified after iteration steps of the ML algorithm. Figure 1, represents the approach in a glance. Spatial tracking phase, however uses the most recent model parameters and updates them. Each sensor is modeled as one neuron, where a subset of the neurons in neural network report their observations to the FC for feature extraction of the spatial signal that finally results in spatial signal recognition. During the iteration steps of the algorithm the FC queries the neural network using the new model parameters, until convergence. Figure 2, illustrates the single layer neural network model of the proposed algorithm.

The proposed algorithm uses two novel forms of stochastic gradient (SG) method for updating the contour level margin (Δ), in each iteration step of the spatial modeling phase. The novelty of the approach is in using SG to tangibly reduce *the cost* of the spatial signal monitoring. The performance evaluation of the algorithm based on extensive simulations show that the proposed algorithm has acceptable modeling error, is reasonably low-cost, and properly converges, in the presence of filtered observation noise. The model and the performance evaluation parameters are listed in Table 1.

The rest of this paper is organized as follows. In the next section the related works will be reviewed. Then, in Section 3, the background of this research will be explained. The proposed ML algorithm will be presented in Section 4. Then the modeling performance, the cost and the convergence of the algorithm will be discussed, based on extensive computer simulations, which is a common approach in evaluation of ML and SG problems.

## 2. Related Works

In this section a number of the related works to this research are reviewed. These researches are categorized in three groups of (*i*) modeling the spatial signal in sensor field using the iso-contour lines of the signal and contour detection problem, (*ii*) using ML for spatial signal recognition, and (*iii*) using SG methods in ML for signal identification.

### 2.1. Spatial Modeling Using Contour Lines

To monitor and track the spatial distribution of temperature, Lian, et al., modeled the spatial signal using its equally spaced contour lines and tracked the changes based on time series analysis [19]. Detection and delineation of the borders of an area, such as the area surrounded by a given contour line, was discussed based on a binary detection measures by Chintalapudi, et al., [20]. Contour detection by clustering in wireless sensor network (WSN) in the presence of observation noise, quantization noise and imperfect radio channel was discussed in [21,22,23,24]. The effect of observation noise and quantization noise for contour detection in WSN using a distributed filter-based approach was discussed in [25]. Based on this filter-based approach, monitoring of a two-dimensional Gaussian signal over time was discussed in [26]. A low-cost protocol was introduced in [27] for detection of iso-contour lines of spatial distribution in WSN. To approximate the iso-contour lines of a given spatial signal, k-nearest neighbors was used in [28,29]. A data-driven distributed algorithm was introduced in [30] to search for the wireless sensors that represent the iso-contours of a spatial signal. A distributed algorithm was introduced in [31] for energy efficient tracking of the iso-contours of a random spatial signal. To find the number of required contour lines and the spacing between the contour levels, an iterative on-demand algorithm was discussed in [32] for spatial signal monitoring in WSN in the presence of observation noise. Spatial signal modeling using its contour lines is comparable with efficient sampling of one-dimensional signals based on level-crossing sampling [33,34]. A novel SG algorithm for low-cost spatial signal monitoring using iso-contours was discussed in [35]. To improve the performance and to resolve the shortcomings of this algorithm, a weighted stochastic gradient (WSG) algorithm was proposed in [36] by adding a weight factor to the gradient term.

### 2.2. Spatiotemporal Recognition Using ML

Once the objective is to represent the whole dynamics of a spatiotemporal signal using finite number of measurements, Gaussian process-based machine learning provides a powerful tool for nonparametric regression and classification [37]. Certain classes of temporal or spatiotemporal Gaussian process regression problems can be converted into finite or infinite dimensional state-space models, where it results in computationally efficient algorithms [37].

Detection of spatiotemporal features of esophageal abnormality from endoscopic videos by incorporating 3D convolutional neural network and convolutional long short-term memories (LSTM) reported in [38] for the first time. Bayesian machine learning (BML) was discussed as a method to extract the electroencephalography (EEG) and magneto-encephalography (MEG) informative brain spatiotemporal–spectral patterns [39].

A hybrid machine learning algorithm was proposed and discussed in [40] in order to minimize and optimize the access time to database for reducing the analysis time and increasing the accuracy of nitrogen vegetation spatiotemporal mapping.

To detect and to visualize the complex behavior in spatiotemporal volumes, a machine learning algorithm has been proposed in [41]. The algorithm detects the spatiotemporal regions of various complexities by training several models.

The spatiotemporal and the steady-state gait pattern of glaucoma patients were studied using body-worn sensors by development of signal processing and machine learning algorithms in [42].

In study of the results from Levodipa challenge on Parkinson’s disease motor symptoms, using the sensor data, spatiotemporal features were calculated. Multiple machine learning methods such as square support vector machine (SVM), decision trees and linear regression were trained to predict the state of the patients [43].

A data-based spatiotemporal modeling method was investigated in [44] for online estimation of temperature distribution in Lithium-Ion batteries in electric vehicles using machine learning algorithm.

An effective spatiotemporal model to predict the temperature distribution in industrial thermal processes was proposed and discussed in [45]. The proposed method showed better performance than that of neural networks and least square SVM.

### 2.3. SG Method Applied in ML Algorithms

For environmental and resource planning, a spatiotemporal planning was proposed based on factored Markov decision process and presents a policy gradient planning to optimize a stochastic spatial policy in [46]. Markov chain Monte Carlo simulation is used to sample landscape policies and estimate their gradients.

A nonparametric feature projection framework was proposed for dimensionality reduction by using mutual information-based stochastic gradient descent in [47].

An iterative algorithm based on stochastic gradient was proposed for cost-efficient monitoring of spatially correlated signals in [35]. An improvement to that algorithm was proposed in [36] using weighted stochastic gradient algorithm for cost-efficient tracking of spatially correlated signals. Later a SG-based ML algorithm was introduced to autonomously identify the model parameters for low-cost spatial tracking of correlated signals in [48]. An accelerated learning algorithm was introduced in [49] to control the iteration pace of the spatial tracking algorithm. This algorithm shows faster convergence in spatial modeling of correlated signals.

## 3. Problem Statement and Background

In this section the technical elements and the background of using SG method as a cost-efficient approach for monitoring of spatially correlated signals is detailed. The distribution of an unknown spatially correlated signal such as g(x,y;t) is assumed over an ad-hoc wireless sensor field. The objective is to monitor this signal in a cost-efficient way over time using sensor observations of a subset of $N_{0}$ wireless sensors that are randomly distributed over the field. It is assumed that $\left(x_{k},y_{k}\right)$, the coordinates of the sensor $S_{k}\;\forall \;k$, is known for the fusion center (FC). The spatial correlation of the unknown signal and the coordinates of the wireless sensors are the only assumptions of this problem.

In WSN, among sensing, computation and communication; communication tangibly consumes most of the in-network energy. Accordingly, in this paper the percentage of sensors in which initiate a communication attempt is taken as cost.

To reduce the spatial monitoring cost, the signal is modeled using its *M* contour lines at levels ${\left\{\ell_{j}\right\}}_{j=1}^{M}$. With this model, the spatial signal is compressed into these *M* contour lines and only those sensors that their local filtered observations $s_{k}\;\forall \;k$, are within the range $\ell_{j}-\Delta \leq s_{k}\leq \ell_{j}+\Delta ,\;\forall \;j,k$, report to the FC, on demand. It is assumed that the sensor observations are polluted with additive white Gaussian noise with zero-mean. To reduce the noise strength, each sensor applies a moving average filter with sufficient window size on its local samples to effectively reduce the noise effect. In reply to the query of the FC, the sensor $S_{k}\;\forall \;k$ reports its observation $s_{k}=g\left(x_{k},y_{k}\right)+z$ to the FC, where *z* is the filtered noise after local moving average filtering. The moving average filter’s window size ($W_{MA}$) can be adjusted, adaptively at each sensor by finding the noise variance, iteratively and empirically, or setup to a known size, based on previously known information. However, in this article, $W_{MA}$ is selected constant and equal to 10, for simplicity. Upon reception of the sensor observations at the FC, a spline interpolation [50] module provides an estimation of the spatial signal. The FC uses the most recent spatial signal estimation to update the contour levels ${\left\{\ell_{j}\right\}}_{j=1}^{M}$. In each iteration and for a finer signal estimation, the FC increases the number of contour levels, *M*. The process of incrementing the number of contour levels continues until convergence of the algorithm. In the course of the signal identification, the FC discovers the signal strength range: $\left(L_{Min},L_{Max}\right)$, its probability density function (PDF): $f_{g}\left(s\right)$, and the spatial and spectral attributes of the spatial distribution.

By modeling the spatial signal using its *M* contour lines and calling for the sensor observations of those sensors that are within Δ margin of the contour levels ${\left\{\ell_{j}\right\}}_{j=1}^{M}$, $N_{r}$ sensors in average will reporting to the FC, according to (Equation 1). Here we assume that the Δ margin of the neighboring contour lines are disjoint.
(1)$$N_{r}=N_{0}\sum_{k=1}^{M}\int_{\ell_{k}-\Delta }^{\ell_{k}+\Delta }f_{g}\left(\gamma \right)d\gamma$$
$N_{r}$ in (Equation 1) is the mathematical expectation of the number of reporting sensors to the FC. Conditionally and when Δ is small enough, (Equation 1) is reduced to (Equation 2).
(2)$$N_{r}≅2N_{0}\Delta \sum_{k=1}^{M}f_{g}\left(\ell_{k}\right)$$

According to (Equation 1) and (Equation 2), $N_{r}$, the expected number of reporting sensors to the FC depends on $N_{0}$, *M*, Δ, as well as the perimeter of the contour lines at each level $\ell_{j}$, j=1,2,…,M. By increasing *M* to get finer signal estimation, the expected value for $N_{r}$ (cost) rises.

When Δ is constant, according to (Equation 1) and (Equation 2), by increasing the number of contour levels (*M*), the number of reporting sensors to the FC increases, where it results in drastic rise in the cost of spatial monitoring. To meet the energy conservation requirements of WSN, a cost efficient approach based on using SG was proposed in [35]. The significance of the proposed stochastic gradient algorithm in [35] is relating the cost of spatiotemporal monitoring to the spatial monitoring performance. During the iterations steps of the stochastic gradient algorithm and as the number of contour levels increases, Δ shrinks, such that at the end, the expected number of reporting wireless sensors to the FC becomes affordable.

By increasing the number of contour levels, the spatial signal estimation error gradually drops. In the proposed SG method in [35], the contour level margin (Δ) is updated related to the slope of the iteration error, and according to (Equation 3). In this equation, the gradient of error is normalized to average of the error strength after [51], to reduce the relevance of Δ to the instantaneous error’s magnitude. In (Equation 3), $\nabla Error_{k-1}=\left(Error_{k-1}-Error_{k-2}\right)$ and ${\bar{Error}}_{k-1}=1/2\left(Error_{k-1}+Error_{k-2}\right)$.
(3)$$\Delta_{k}=\Delta_{k-1}\left(1+\frac{\nabla Error_{k-1}}{2{\bar{Error}}_{k-1}}\right)$$

Because the actual spatial signal g(x,y) is unknown to the FC, instead of spatial signal estimation error, iteration error is used in calculation of the gradient. Iteration error is defined according to (Equation 4). The simulation results showed that the iteration error behaves very noisy in comparison to spatial signal mean absolute error (MAE), which is defined according to (Equation 5) [36].
(4)$$Error_{k}=\sum_{i=1}^{P}\sum_{j=1}^{Q}\frac{\left|{\tilde{g}}_{k}\left(x_{i},\;y_{j}\right)-{\tilde{g}}_{k-1}\left(x_{i},\;y_{j}\right)\right|}{P\times Q}$$
(5)$$MAE_{k}=\sum_{i=1}^{P}\sum_{j=1}^{Q}\frac{\left|g\left(x_{i},\;y_{j}\right)-{\tilde{g}}_{k}\left(x_{i},\;y_{j}\right)\right|}{P\times Q}$$

In (Equation 4) and (Equation 5), ${\tilde{g}}_{k}\left(x_{i},\;y_{j}\right)$ is the spatial signal reconstruction from the reported sensor observations in the kth iteration at grid point coordinate $\left(x_{i},\;y_{j}\right)$ of the sensor field. The iteration error and MAE in (Equation 4) and (Equation 5) are calculated at P×Q grid points of the sensor field. In calculation of the iteration error and reconstruction error in the paper, we use mean absolute error (norm-1), instead of mean square error (norm-2), because norm-1 does not magnify the relatively large errors in the borders of the sensor field. The large errors in the borders of the sensor field are not recoverable due to sensor selection limitation in the borderline, therefore its large residual error does not allow to properly shrink Δ in (Equation 3). Accordingly, norm-1 results in smaller monitoring cost and better spatial modeling error, in comparison to norm-2 [52].

In selection of the contour levels, equally spaced and optimally spaced contour lines were considered in [35]. The optimally spaced contour lines were selected based on Lloyd-Max algorithm [53], according to (Equation 6) and (Equation 7).
(6)$$\ell_{i}=\frac{\int_{y_{i}}^{y_{i+1}}xf_{g}\left(x\right)dx}{\int_{y_{i}}^{y_{i+1}}f_{g}\left(x\right)dx},\;i=1,2,\ldots ,M$$
where $y_{i}$ is calculated according to (Equation 7).
(7)$$y_{i}=\frac{\ell_{i}+\ell_{i-1}}{2},\;i=1,2,\ldots ,M-1$$

The spatial signal monitoring based on modeling with optimally spaced contour lines outperforms that of the equally spaced contour lines, provided that the PDF of the signal strength, $f_{g}\left(s\right)$ is perfectly known. However, because this PDF is unknown, in this paper we use optimally spaced contour levels only as benchmark to compare the performance of spatial monitoring using equally spaced contour lines.

Even though the proposed approach for spatial monitoring in [35] is low cost, however it does not guarantee that the iterative algorithm meets the monitoring performance of the benchmark. A cost-efficient weighted SG (WSG) algorithm was proposed in [36] to meet the performance of the benchmark. The proposed weighted stochastic gradient algorithm trades-off between the cost and the monitoring performance. In WSG, a constant weight factor 0≤μ≤1 was added to the normalized gradient term, according to (Equation 8).
(8)$$\Delta_{k}=\Delta_{k-1}\left(1+\mu \frac{\nabla Error_{k-1}}{2{\bar{Error}}_{k-1}}\right)$$

The performance evaluation of the WSG algorithm showed that it outperforms the performance of SG, at no tangible additional monitoring cost [36], if a proper μ is selected. Extensive performance evaluations using computer simulations showed that WSG samples the spatial signals related to their rate of spatial variations. This result supports the sampling theorem requirement that signals with wide bandwidth need to be sampled at higher rate than that of signals with narrow bandwidth. Also, the performance evaluation result of WSG showed that the spatial monitoring algorithm converges better than SG [36].

Even though using WSG algorithm for spatial monitoring of signals has promising outcomes, however searching for the initial factors such as μ, the initial value of Δ and also the signal strength range can be cumbersome and this encourages to adapt a ML algorithm to find the model parameters.

Two update regimes are introduced in the next section to automatically find the model parameters during the iteration steps of the algorithm. Instead of a constant weight factor μ, two variable gain stochastic gradient approaches are introduced. The performance evaluations show that the proposed approaches are low-cost, converge to nearly the same model parameters, and have low sensitivity to noise than that of WSG and SG.

## 4. The Proposed Algorithms

The proposed algorithm in [36] improved the performance of the stochastic gradient algorithm that was introduced in [35], by adding a constant weight factor μ in updating the value of Δ, according to (Equation 8). The study of the convergence of the signal strength range in spatial monitoring using WSG [36] showed that it smoothly converges toward the actual signal strength range within a few iteration steps. However finding a proper value of μ needs extensive initial search. Here, we use this convergence behavior to create a replacement for the weight factor μ. In this section, two different adjustment methods are proposed to update weight factor in successive iterations of the algorithm to assign a final value to Δ. The proposed weight adjustment methods nearly converge to the same final value of Δ, and according (Equation 1) it is expected to have the same tracking cost.

To identify the spatial signal, the sensor observations of selected subsets of wireless sensors are iteratively used to reconstruct the spatial signal at the FC. The algorithm finds the model parameters, such as Δ, *M*, and ${\left\{\ell_{j}\right\}}_{j=1}^{M}$. Here we use stochastic gradient method with adapted parameters to identify the model parameters, automatically. Study of the convergence of the signal strength range in spatial monitoring using WSG [36] showed that it smoothly converges to the actual signal strength range within a few iteration steps. Here, we use this convergence behavior to create a replacement for the weight factor. In a general trend, in the kth iteration step of the algorithm, the detected signal strength $R_{k}=\left(L_{min,k},L_{max,k}\right)$ becomes closer to the actual signal strength range, where $L_{min,k}$ and $L_{max,k}$ are the minimum and the maximum of the spatial signal strength in the kth iteration, after spline interpolation at the FC, respectively. Here we define the signal strength range-difference as: $RD_{k}=L_{max,k}-L_{min,k}$. The ratio of two successive $RD_{k}$ is define as the signal strength range span ratio (SRSR) according to (Equation 9). It is expected that during the iterations steps of the algorithm, $SRSR_{k}$ first approaches to the neighborhood of 1.0 and based on the residual noise in sensor observations, fluctuates around 1.0.
(9)$$SRSR_{k}=\frac{RD_{k-1}}{RD_{k}}$$

Now, we use $SRSR_{k}$ to modify (Equation 3) and (Equation 8) and to introduce methods that automatically initiate and update the model parameters until convergence. We call the first method *adaptive weight stochastic gradient (AWSG)* and the second method *scaled stochastic gradient (SSG)*. These two weight factors were obtained by experiments and after observation of the variation of $SRSR_{k}$ in the proposed algorithm in [36]. The performance evaluation results that are given in the next section show that with these changes the algorithm converges faster than that of [36], it does not need manual setup for μ and also it becomes independent from the initial guess for $\Delta_{0}$.


***Adaptive Weight Stochastic Gradient (AWSG):***


In *AWSG*, we replace μ in (Equation 8) with a function of $SRSR_{k}$. With this change the update equation for Δ changes according to (Equation 10).
(10)$$\Delta_{k}=\Delta_{k-1}\left(1+\mu_{k}\;\frac{\nabla Error_{k-1}}{2{\bar{Error}}_{k-1}}\right)$$
where $\mu_{k}$ is according to (Equation 11).
(11)$$\mu_{k}=\sqrt{\frac{SRSR_{k-1}}{1+SRSR_{k-1}}}$$

***Scaled Stochastic Gradient (SSG):***   

In the second method, *SSG*, the modification factor is applied to (Equation 3), instead of (Equation 8). With this change, (Equation 3) is modified to (Equation 12).
(12)$$\Delta_{k}=SRSR_{k-1}\;\;\Delta_{k-1}\left(1+\frac{\nabla Error_{k-1}}{2{\bar{Error}}_{k-1}}\right)$$

Besides faster convergence, the recent two changes also help avoid the shrinkage of Δ faster than that it fails to continue spatial monitoring due to lack of enough reporting sensor observations.

As previously mentioned, the algorithm is implemented in two phases of spatial modeling, where the model parameters such as Δ and ${\left\{\ell_{j}\right\}}_{j=1}^{M}$ are selected; and spatial tracking where the algorithm uses the same Δ and *M*, and it updates the new contour levels ${\left\{\ell_{j}\right\}}_{j=1}^{M}$. The spatial modeling phase continues until convergence of Δ and the signal strength range. During the spatial tracking, *only those sensors* that their observations are within the Δ margin of the contour levels are queried. Accordingly, the spatial tracking has relatively small cost.

***Implementation of the Spatial Modeling Phase***:

In implementation of the algorithm and for *initiation* of the spatial modeling, the FC sends queries to two small groups of randomly picked sensors from the sensor field. The average of the sensor observations of these two groups form the $L_{min}$ and $L_{max}$. To accelerate the process and to reduce the number of iterations, the number of contour levels are incremented for κ=3.

In *step two*, the FC selects an initial value for $M_{0}$, between 3 and 10, then finds the initial, equally spaced contour levels ${\left\{\ell_{k}\right\}}_{k=1}^{M_{0}}$ between $L_{min}$ and $L_{max}$, and the initial value for $\Delta =(\ell_{2}-\ell_{1})/2$.

Then, in *step three*, the FC queries the sensor field by sending the ${\left\{\ell_{k}\right\}}_{k=1}^{M_{0}}$ contour levels and $\Delta_{0}$, and requests for the reply of those sensors that their sensor observations are within the range $\ell_{j}-\Delta_{0}\leq s_{k}\leq \ell_{j}+\Delta_{0}\;,\;\forall \;j,k$.

In *step four*, after receiving the query replies from the sensor field, the spatial signal is reconstructed at the FC from the sensor observations.

Next, in *step five*, the new signal strength range ($L_{min},L_{max}$) is found from the reconstructed signal, $M=M_{1}\leftarrow M_{0}+\kappa$, the new contour levels ${\left\{\ell_{k}\right\}}_{k=1}^{M_{1}}$ are calculated, and the new Δ is $\Delta_{1}=\Delta_{0}$. Here, κ=3.

In *step six*, the FC queries the sensor field by broadcasting $M_{1}$ new contour levels and $\Delta_{1}$.

In *step seven*, after receiving the query replies from the sensor field, the FC attempts the spatial signal reconstruction.

Then, in *step eight*, the FC updates the value of Δ from Equations (Equation 10) or (Equation 12).

In *step nine*, the FC calculates the new number of contour levels M←M+κ and their levels ${\left\{\ell_{k}\right\}}_{k=1}^{M}$.

In *step ten*, the FC queries the sensor field by broadcasting the new contour levels ${\left\{\ell_{k}\right\}}_{k=1}^{M}$ and requesting the sensor observations of those sensors that falls within range $\ell_{j}-\Delta_{n}\leq s_{k}\leq \ell_{j}+\Delta_{n}\;,\;\forall \;j,k$. Similarly, κ=3.

Then the FC repeats the process from *step seven*, until convergence.

The summary of the discussed algorithm for spatial modeling phase is presented in Table 2.


***Spatial Tracking Phase:***


After convergence of the algorithm in spatial modeling phase, the FC uses the same final Δ and the same number of contour levels *M* at convergence, and just updates the contour level set ${\left\{\ell_{k}\right\}}_{k=1}^{M}$, on demand.

## 5. Performance Evaluation

For performance evaluation of the proposed ML algorithm, first we introduce the spatial signal construction model and also the simulation assumptions. Then, in the next part of this section, the performance evaluation results will be given.

### 5.1. Spatial Signal Model and Assumptions

To construct the spatial signal, similar to [35,36], diffusion model is used to synthesize the spatial signal. The reasons for using this model are its simplicity and capability to analytically change the spatial signal in performance evaluation of the algorithm in spatial tracking. The diffusion model was introduced to model the correlated spatial signals [54].

In the proposed ad-hoc WSN problem, the wireless sensor nodes are assumed randomly distributed, with Poisson distribution over a known area *A* with dimensions of 100 × 100. It is assumed that the network is localized, meaning that the FC knows the coordinates of the wireless sensors. Each and every of the wireless sensors in the sensor field can communicate with the FC, either by multi-hopping or by direct communication. For performance evaluation of the proposed algorithm, we used MATLAB. The related simulation codes are available online for verification purpose [55].

The correlated spatial signal g(x,y) is analytically formed using (Equation 13). According to this approach the synthetic signal is formed by superposition of a large number of two-dimensional Gaussian distributions G(mx,my,σ), that are randomly distributed at center points (mx,my), over the sensor field, each formed according to (Equation 14).
(13)$$g\left(x,y\right)=\sum_{p=1}^{N_{1}}a_{p}G\left(mx_{p},my_{p},\sigma_{1}\right)+\sum_{p=1}^{N_{2}}b_{p}G\left({\hat{mx}}_{p},{\hat{my}}_{p},\sigma_{2}\right)$$
(14)$$G\left(mx,my,\sigma \right)=exp\left(-\frac{{\left(x-mx\right)}^{2}+{\left(y-my\right)}^{2}}{2\sigma^{2}}\right)$$

The synthetic spatial signal is formed by summation of two groups of Gaussian distributions with two different standard deviations of $\sigma =\sigma_{1}$ or $\sigma_{2}$, as it is detailed in (Equation 13). The coefficients $a_{p}$ and $b_{p}$ in (Equation 13) are random positive weight factors for the spatial Gaussian signals, so that the final synthetic signal is limited inside a range (0,100). Figure 3, illustrates an instance of the synthetic spatial signal constructed using (Equation 13). For generation of this spatial signal, $\sigma_{1}$ and $\sigma_{2}$ are assumed equal to 5 and 10, respectively.

For this performance evaluation we assumed either 10,000 or 12,000 wireless sensors in the sensor field. The MAE of the proposed algorithm, its cost and also the convergence of the algorithm are investigated based on extensive computer simulations. As benchmark and for comparison, similar to [35,36], spatial modeling is used with optimal contour levels, based on Lloyd-Max, according to (Equation 6) and (Equation 7), when the PDF of the signal strength is assumed.

### 5.2. Performance Evaluation Results


***Spatial Modeling MAE:***


The spatial modeling MAE of the proposed ML algorithm for modeling of Figure 3 at different noise strengths is illustrated in Figure 4 and Figure 5. Figure 4, compares the spatial modeling of AWSG, WSG and benchmark (Lloyd-Max). As this figure illustrates, AWSG, similar to WSG [36], converges to the modeling performance of benchmark, at the same observation noise strength. According to this result, AWSG converges a bit faster than that of WSG. Figure 5, compares the convergence of AWSG and SSG, where it shows that SSG converges in most of the cases slightly faster than AWSG. In this paper, the spatial modeling errors (MAE and RMSE) are sketched in dB, $MAE_{k}\left(dB\right)=20Log_{10}AME_{k},\forall k$.

The modeling performance of AWSG using norm-2, which is defined in (Equation 15), is compared with norm-1 (MAE), in Figure 6. According to this result, the modeling performance using norm-1 (MAE) is around 3 dB better than that of norm-2, which is root mean square error (RMSE). Norm-2 results in poorer modeling performance, because it magnifies the large error spikes that usually happens in borderlines, where the sensor population is limited.
(15)$$RMSE_{k}=\sqrt{\sum_{i=1}^{P}\sum_{j=1}^{Q}\frac{{\left[g\left(x_{i},\;y_{j}\right)-{\tilde{g}}_{k}\left(x_{i},\;y_{j}\right)\right]}^{2}}{P\times Q}}$$


***Spatial Modeling Cost:***


The relative spatial modeling cost of the proposed ML algorithm for AWSG and SSG is presented in Figure 7. The relative spatial modeling cost is calculated based on $R=Cost_{SSG}\left(k\right)/Cost_{AWSG}\left(k\right)$. According to this figure, AWSG has relatively smaller spatial modeling cost than that of SSG. Also as this figure shows, by increasing the standard deviation of noise from σ=0.1 to 0.3, the cost of the algorithm (the percentage of the reporting wireless sensor nodes to the FC), increases, due to increase in false detections. Increase in the number of false detection due to the presence of observation noise has been addressed in other literature, as well [25].


***Spatial Tracking Cost:***


The spatial tracking cost of the proposed ML algorithm using SSG is illustrated in Figure 8. According to this figure, as the observation noise’s strength increases, the tracking cost of the algorithm, rises. However, the percentage of reporting sensors to the FC is maintained around 10% or less, when the standard deviation of the additive filtered noise is below 0.5. According to (Equation 1), the number of reporting sensors to the FC depends on Δ. Later in this section we show that the final Δ for SSG and AWSG is almost the same. Accordingly, we expect that the cost of AWSG and SSG be nearly the same. Evaluations based on computer simulations proves this expectation.

The effect of sensor population on spatial modeling’s cost using AWSG is illustrated in Figure 9. The results of this figure are related to observation noise of $\sigma_{Noise}=0.3$ and two sensor population of 10,000 and 12,000. According to this figure, with 10,000 sensors in the sensor field, around 8.5% of the sensors (around 850 sensors), report to the FC. Once the sensor population increases to 12,000 sensors, the total reporting sensors is 7.45% (around 895 sensors). This outcome that was also reported for WSG in [36], states that by increasing the number of sensors in the sensor field, the cost of the algorithm does not tangibly rise.


***Spatial Tracking MAE:***


The spatial modeling AME of AWSG at different observation noise’s strength are illustrated in Figure 10. A comparison between the results of this figure and Figure 4 and Figure 5, clears that the spatial tracking MAE of the proposed tracking algorithm is about the same as that of spatial modeling MAE after convergence. Therefore, using the final spatial modeling parameters for tracking is a tractable approach. According to Figure 10, as the spatial signal changes due to its temporal variations, the spatial tracking MAE slightly increases. As result, in the course of time the model gradually becomes poorer and another round of spatial modeling will be required.


***Convergence of*Δ:**


The variation of Δ in the course of the convergence for AWSG and SSG is illustrated in Figure 11. According to this figure, in SSG Δ sharply rises first and then aggressively drops until convergence. This figure also shows that Δ in AWSG gradually drops in the course of the convergence of the algorithm. This figure shows that both SSG and AWSG converge to around the same final values. According to this figure, and based on (Equation 1), it is expected that the tracking cost of AWSG and SSG be around the same value. Also, due to the sharp rise in Δ in SSG, the relative cost of AWSG over SSG drops suddenly. This fact is illustrated in Figure 7. The results of Figure 11 also show that Δ is slightly sensitive to observation noise’s strength.


***Signal Strength Range Span Ratio (SRSR):***


Convergence of $SRSR_{k}\;,\;\forall k$ during the spatial modeling process is the other factor, which is used in development of AWSG and SSG in (Equation 10) and (Equation 12), respectively. SRSR, which is defined in (Equation 6) converges to around 1.0, in the course of spatial modeling. This convergence is illustrated in Figure 12. According to this figure, the presence of noise results in some final misadjustment and noisy variation around 1.0. By increase of the noise strength in sensor observations the misadjustment is expected to rise.

## 6. Conclusions

Two machine learning (ML) approaches based on adaptive weight stochastic gradient (AWSG) and scaled stochastic gradient (SSG) are proposed and discussed for spatial signal modeling and tracking. The spatial signal, which is monitored using sensor observations, is modeled with a number of its contour lines at equally spaced levels. The fusion center (FC) calls for a subset of sensor observations that fall within a given Δ margin of each of these contour levels. The ML algorithm iteratively varies the number of contour levels and Δ until convergence. Convergence of Δ and the signal strength’s range are two measures that can be used for convergence of the algorithm. Performance evaluations based on computer simulations show that the proposed algorithms have relatively the same cost and modeling/tracking performance. The tracking cost of the algorithm is around 10% or less, when the filtered observation noise’s standard deviation is around 0.5 or less. With the increase of observation noise strength, the cost also increases. The number of reporting sensors to the FC remains almost the same, when the sensor population increases. The tracking MAE of AWSG and SSG are around the spatial modeling MAE, in the same noise strength.

## Figures and Tables

**Figure 1 sensors-21-05175-f001:**
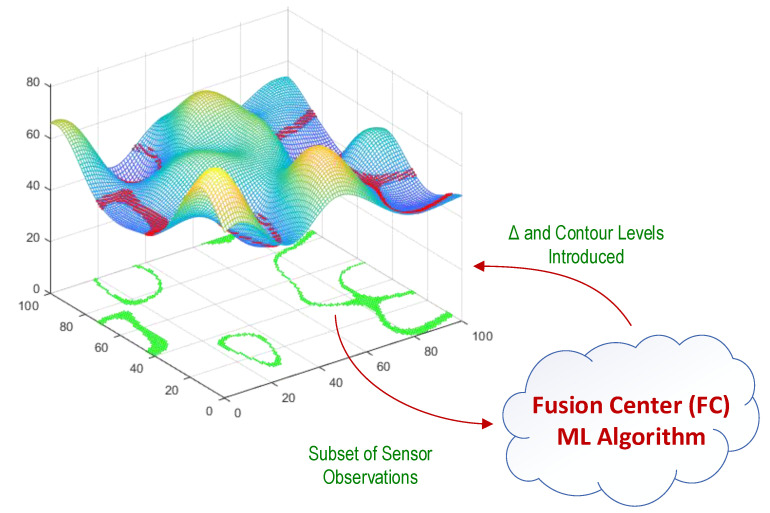
A subset of sensor observations are reported in response to the FC’s query to reconstruct the signal. The FC uses these sensor observations for feature extraction and reconstruction of the spatial signal. This figure illustrates a given subset of wireless sensors with green color and their related sensor observations in margin of a given contour level, with red colors on spatial signal distribution.

**Figure 2 sensors-21-05175-f002:**
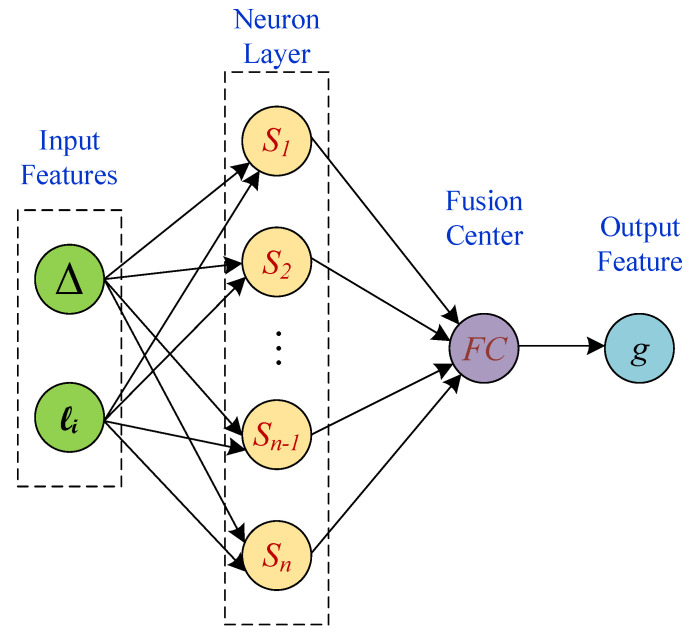
A subset of the sensors that are modeled as neurons report to the FC for feature extraction purpose.

**Figure 3 sensors-21-05175-f003:**
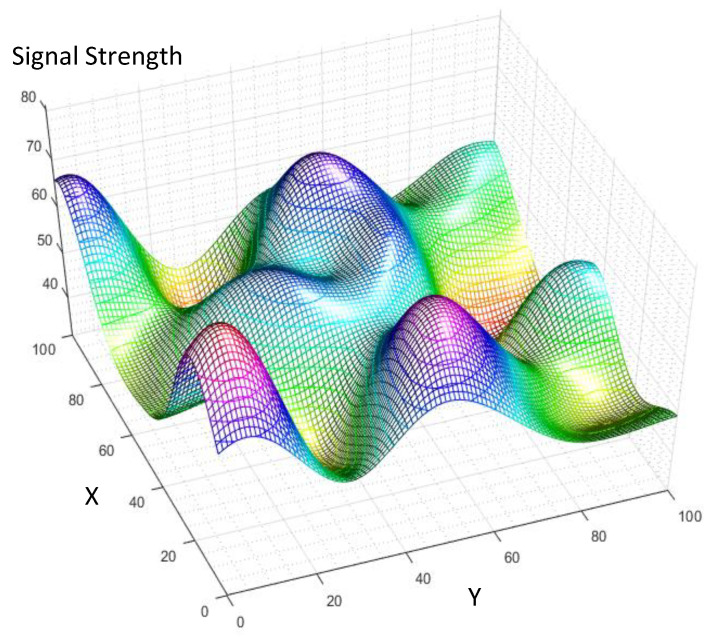
The synthetic correlated spatial signal is generated using the described model in (Equation 13).

**Figure 4 sensors-21-05175-f004:**
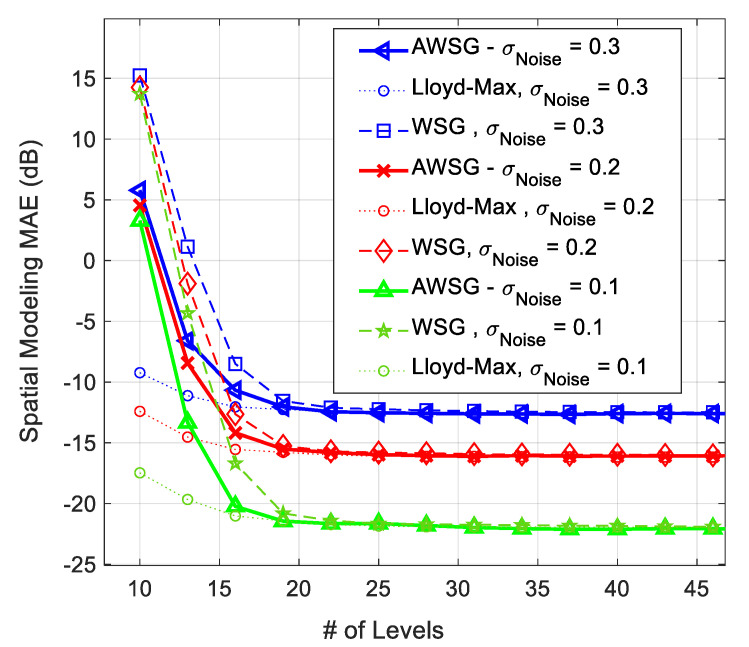
Comparison between the spatial modeling MAE of AWSG and WSG in the presence of different noise strengths $\sigma_{N}$, during convergence process of the algorithm.

**Figure 5 sensors-21-05175-f005:**
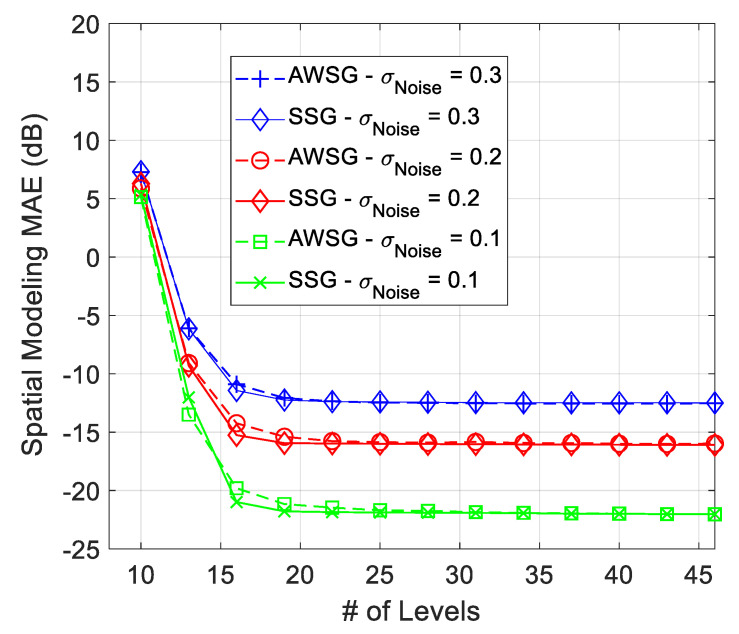
Comparison between the convergence speed of AWSG and SSG, during convergence process of the algorithm.

**Figure 6 sensors-21-05175-f006:**
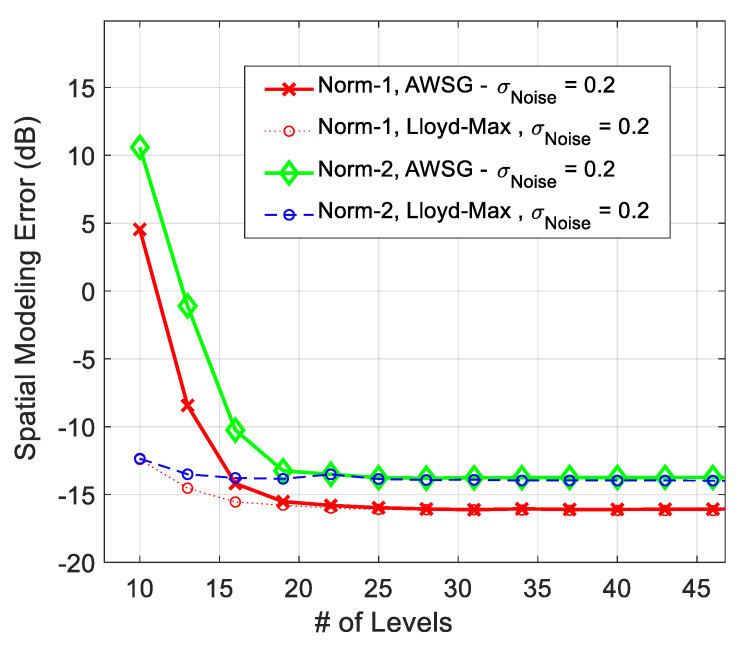
Comparison between the spatial modeling error of norm-1 (MAE) and norm-2 (RMSE) in convergence process of AWSG.

**Figure 7 sensors-21-05175-f007:**
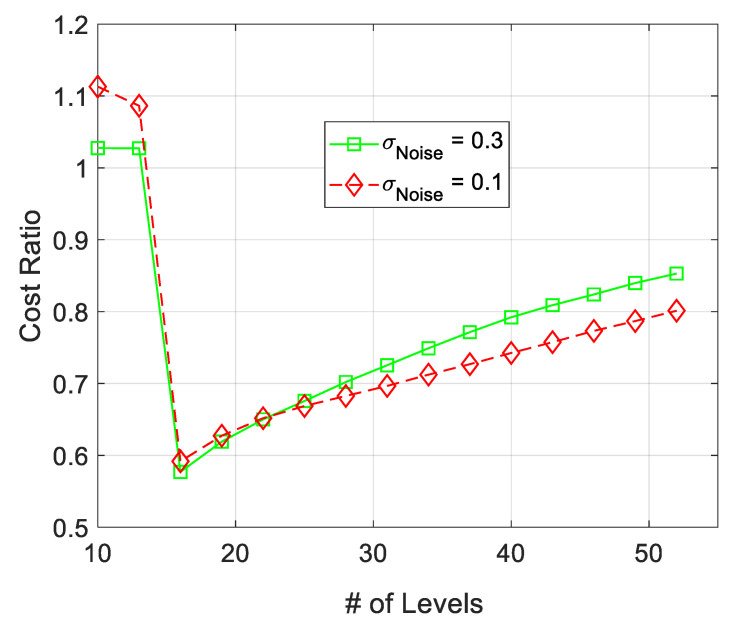
Comparison between the spatial modeling cost of AWSG and SSG. The relative cost $R=Cost_{SSG}\left(k\right)/Cost_{AWSG}\left(k\right)$ is sketched for two different filtered noise strength in sensor observations.

**Figure 8 sensors-21-05175-f008:**
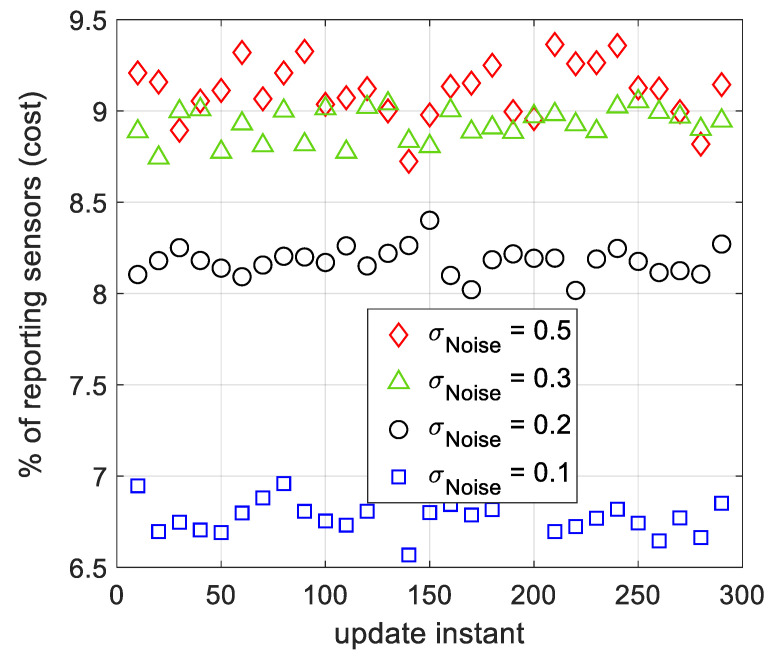
The tracking cost of SSG for several filtered observation noise strengths, during several tracking instances. The sensors apply moving average filter on their local observations to reduce the effect of additive noise.

**Figure 9 sensors-21-05175-f009:**
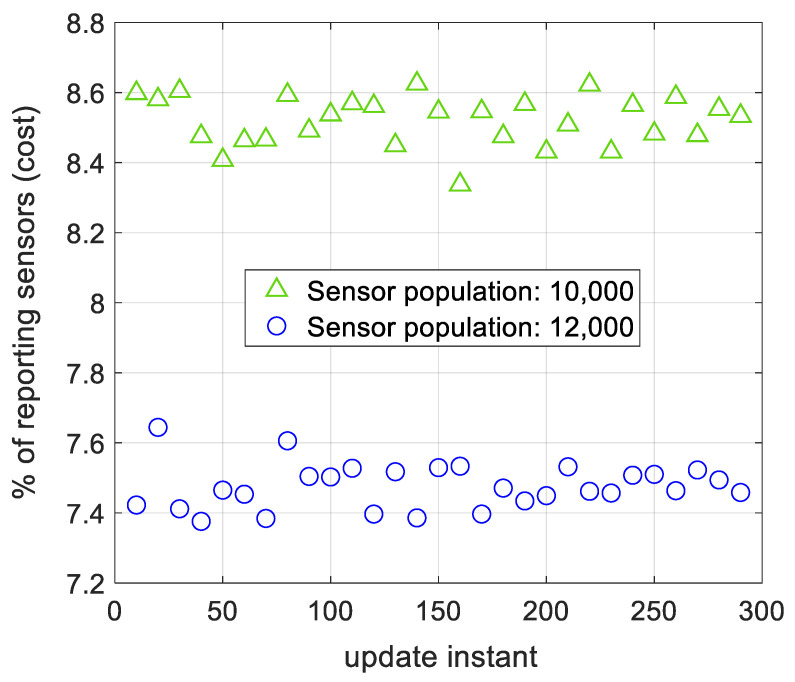
The percentage of reporting sensors to the FC for two different sensor populations. The number of reporting sensors does not tangibly change, when the sensor population changes.

**Figure 10 sensors-21-05175-f010:**
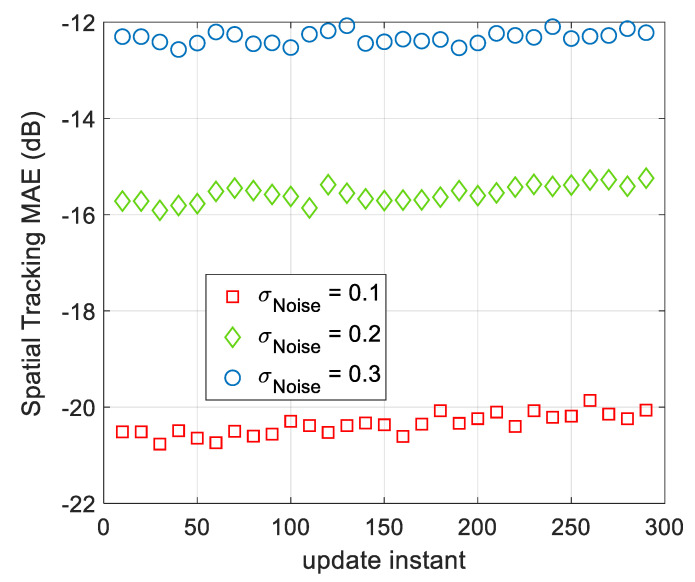
The spatial tracking MAE of AWSG at different observation noise’s strength. As the standard deviation of the observation noise increases, the modeling performance drops.

**Figure 11 sensors-21-05175-f011:**
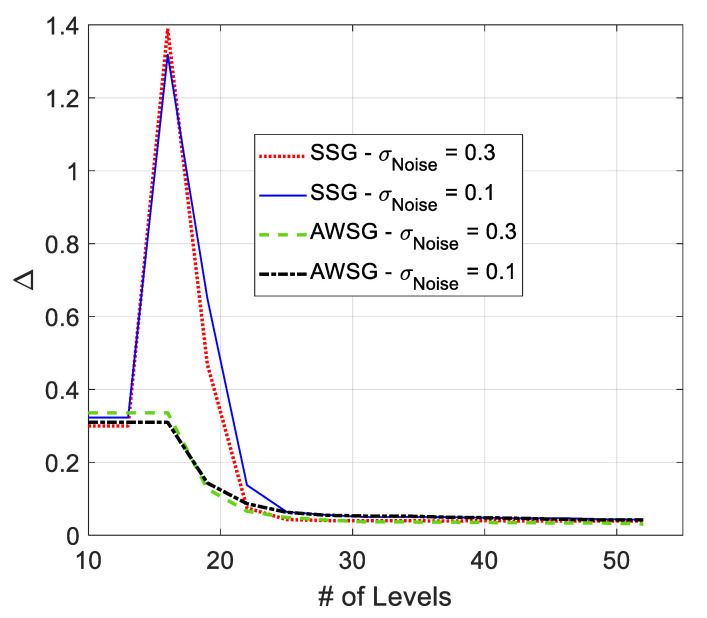
Variation of Δ in the course of convergence in SSG and AWSG in the presence of filtered observation noise.

**Figure 12 sensors-21-05175-f012:**
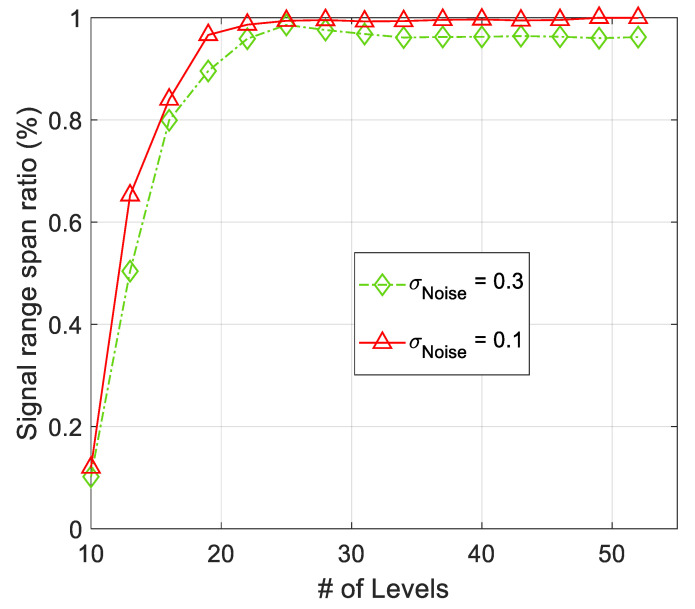
Convergence of *SRSR* during the iteration steps of the algorithm. The effect of observation noise is some misadjustment in value of *SRSR* after convergence.

**Table 1 sensors-21-05175-t001:** Parameters of the proposed algorithm.

Parameter	Description	Selection
*M*	The number of contour levels	Adaptive
${\left\{\ell_{j}\right\}}_{j=1}^{M}$	Contour level set	Adaptive
Δ	Contour line’s margin	Adaptive by ML
$L_{min}$	Reported lower signal strength	Most recent search
$L_{max}$	Reported upper signal strength	Most recent search
$N_{0}$	Number of the wireless sensors in the field	10,000 or 12,000
$M_{0}$	Initial number of iso-contour lines	Initial guess (3 $\leq M_{0}\leq$ 10)
$\sigma_{Noise}$	Noise’s standard deviation after averaging	——-
κ	Increment in the number of contour lines	κ is selected 3
$\mu_{k}$	The stochastic gradient weight factor	Adaptive
$x_{step}$ $y_{step}$	The horizontal and vertical shifts of the signal elements	0.1 per time step
$W_{MA}$	Window size of the moving average filter	Adaptive or fixed (10)

**Table 2 sensors-21-05175-t002:** Summary of the proposed algorithm.

1. The FC queries two small sets of sensors at random locations for the signal strength range $\left(L_{min},L_{max}\right)$. 2. With initial $M=M_{0}$ contour levels, the FC finds ${\left\{\ell_{i}\right\}}_{i=1}^{M_{0}}$ in $\left(L_{max},L_{min}\right)$ and $\Delta_{0}=\left(\ell_{2}-\ell_{1}\right)/2$. 3. The FC queries the sensor field with the ${\left\{\ell_{i}\right\}}_{i=1}^{M_{0}}$ and $\Delta =\Delta_{0}$. 4. The FC receives those sensor observations that are in Δ margin of the contour levels and reconstructs the spatial signal. 5. The FC updates the signal strength range $\left(L_{min},L_{max}\right)$ and $M=M_{1}\leftarrow M_{0}+\kappa$ and uses the same $\Delta_{1}=\Delta_{0}$. 6. The FC queries the sensor field with the new *M*, ${\left\{\ell_{i}\right\}}_{i=1}^{M_{1}}$ and Δ. 7. The FC receives the query replies, reconstructs the signal, find the new signal strength range. 8. The FC finds the new Δ according to either (Equation 10) or (Equation 12). 9. The FC finds the new M←M+κ and the set of equally spaced levels ${\left\{\ell_{i}\right\}}_{i=1}^{M}$. 10. Th FC queries the sensor network with the new *M*, ${\left\{\ell_{i}\right\}}_{i=1}^{M}$ and Δ. 11. Repeat from Step (7), until convergence.

## Data Availability

Not applicable.

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
