# Peer review of "Communication-Efficient Tracking of Unknown, Spatially Correlated Signals in Ad-Hoc Wireless Sensor Networks: Two Machine Learning Approaches"

_sensors, 2021, doi:10.3390/s21155175_

Round 1

Reviewer 1 Report

A low-cost machine learning (ML) algorithm is proposed and discussed for spatial tracking of unknown related signals in local, ad-hoc wireless sensor networks. Each sensor is modeled as a neuron, and a selected subset of these neurons is called to identify spatial signals. The algorithm is implemented in two stages: space modeling and space tracking. The spatial signal is modeled using different contour levels, and those sensors within a set variation threshold of any one of these levels report their sensor observations to the fusion center (FC) to obtain spatial signal reconstruction.

The spatial modeling stage of the algorithm determines the number of these contours, their levels and the appropriate thresholds. At this stage, the algorithm can use adaptive weight stochastic gradient or scaled stochastic gradient method to select the appropriate threshold. Along with sensor observations, additive white Gaussian noise (AWGN) with zero mean is assumed.  In order to reduce the influence of observation noise, each sensor applies a moving average filter to its observation for noise reduction. Based on computer simulation, the modeling performance, cost and algorithm convergence are discussed. The percentage of wireless sensor communication attempts is assumed to be the cost.

The ML algorithm iteratively varies the number of contour levels and the variation threshold until convergence. Convergence of the variation threshold and the signal strength’s range are two measures used for convergence of the algorithm. Performance evaluations based on Matlab simulations exhibit that the proposed algorithm has relatively the similar cost, in addition to modelling and tracking performance. The tracking cost of the algorithm is around 10% or less.

This algorithm is proposed for application in environmental monitoring. The paper is well organized and presented. The introduction part provides sufficient background information for readers. The problem statement is clearly formulated and the approaches for solving the problem are described in appropriate details. The illustrations and charts also help readers to appreciate the performance of the proposed algorithm. The performance of the algorithm is good as well. However, the author needs to proofread the manuscript further to reduce some typo or grammar mistakes. E.g.,

1) Where is the subject in line 21 “paper presents a machine learning (ML) algorithm for recognition and low-cost”?

2) In line 316, “benchmark and for comparison, similar to [35] and [36], we use spatial modeling with” would better be changed to “benchmark and for comparison, similar to [35] and [36], spatial modeling is used with”.

Author Response

Date: 07/24/2021

Subject: Revised Manuscript ID: sensors-1288639

ATTN: Esteemed Reviewer #1
Journal of MDPI Sensors

Attached, please find the revised paper entitled “Communication-Efficient Tracking Of Unknown, Spatially Correlated Signals In Ad-Hoc Wireless Sensor Networks: Two Machine Learning Approaches”, along with the changes made in the revised manuscript to be considered for publication in your esteemed Journal.

The author wishes to express his most profound gratitude to the Editor-in-Chief, Associate Editor, and the anonymous potential Reviewers for providing all the useful suggestions and constructive criticisms which have significantly improved the quality of this manuscript. The author has addressed all the issues and queries raised by the Editor-in-Chief, Associate Editor, and the Reviewers in the revised manuscript. The author has thoroughly revised the manuscript to improve the quality of the paper. This is my original, unpublished work and has not been submitted to anywhere for reviews. Kindly inform me about the result once the review process will be over.

Special regards,

 Hadi Alasti

Assistant Professor

School of Polytechnic – Purdue University Fort Wayne, IN

Phone: +1 (260) 481-6423

Reply to the Esteemed Reviewer #1

Comment #1 The author needs to proofread the manuscript further to reduce some typo or grammar mistakes.

Respond to Comment #1: I really appreciate you for supportive comments and also for this opportunity to revise and resubmit the manuscript of this research. I worked on the manuscript during the past week to enhance the manuscript and to include the comments of the respected Reviewers.

Comment #2:  Where is the subject in line 21 “paper presents a machine learning (ML) algorithm for recognition and low-cost”?

Respond to Comment #2: I appreciate you for your constructive comment. The missing subject of the sentence in line 21, added.

Comment #3: In line 316, “benchmark and for comparison, similar to [35] and [36], we use spatial modeling with” would better be changed to “benchmark and for comparison, similar to [35] and [36], spatial modeling is used with”.

Respond to Comment #3: I appreciate you for your correcting comment. The sentence was corrected according to your comment.

Reviewer 2 Report

This paper presented a method to track unknown spatially-correlated signals over time using wireless ad-hoc networks.  It was generally well-written except there were occasional English errors throughout, the most obvious of which was the first word of the paper, which was missing, namely , "This".  Also throughout the paper, there were missing qualifiers, e.g., "the", "a", etc. The author should go through the paper carefully and correct any such errors.

The presented method itself was interesting in that it tried to minimize both the numbers of sensor nodes that would have to report to the Fusion Centre, as well as the error based on residual noise at sensor centers.  There were some details that were missing, for example, the value of the window size for the moving average filter (Wma), which was given in the table of parameters, is not given and it is not clear whether this parameter is estimated from the data or is set in an ad-hoc manner.  It is also not clear whether this parameter is based on the noise level or is fixed.  

If the above points are addressed, I believe this manuscript is worthy of publication since it presents a practical and precise method for estimating spatially-coherent signals with ad-hoc wireless sensor networks.

Author Response

(The authors gave the same response as above.)

Reviewer 3 Report

The author presents am improved algorithms to track unknown and spatially correlated signals in ad-hoc wireless sensor networks. Overall, the paper is well written and the contribution to the state-of-the-art in the area is clear. Nevertheless, I have two suggestions that I think could improve the final manuscript. 

The first is to include a figure in the introduction explaining the proposed approach. Sometimes, a visual representation may be very helpful. 

I suggest also including a list with the simulation parameter to facilitate the replication of the results. 

Finally, I identified that Figure 2 does not have any label. Could you include them?

Author Response

Date: 07/24/2021

Subject: Revised Manuscript ID: sensors-1288639

ATTN: Esteemed Reviewer #3
Journal of MDPI Sensors

Attached, please find the revised paper entitled “Communication-Efficient Tracking Of Unknown, Spatially Correlated Signals In Ad-Hoc Wireless Sensor Networks: Two Machine Learning Approaches”, along with the changes made in the revised manuscript to be considered for publication in your esteemed Journal.

The author wishes to express his most profound gratitude to the Editor-in-Chief, Associate Editor, and the anonymous potential Reviewers for providing all the useful suggestions and constructive criticisms which have significantly improved the quality of this manuscript. The author has addressed all the issues and queries raised by the Editor-in-Chief, Associate Editor, and the Reviewers in the revised manuscript. The author has thoroughly revised the manuscript to improve the quality of the paper. This is my original, unpublished work and has not been submitted to anywhere for reviews. Kindly inform me about the result once the review process will be over.

Special regards,

Hadi Alasti

Assistant Professor

School of Polytechnic – Purdue University Fort Wayne, IN

Phone: +1 (260) 481-6423

Reply to the Esteemed Reviewer #3

Comment #1 The first is to include a figure in the introduction explaining the proposed approach. Sometimes, a visual representation may be very helpful.

Respond to Comment #1: I appreciate you for your correcting comment. Figure 1, was added to the early pages of the article to provide a graphical view to the spatial modeling and iterative feedbacks between the sensor field and the fusion center (FC) in the process of machine learning algorithm.

Comment #2:  I suggest also including a list with the simulation parameter to facilitate the replication of the results.

Respond to Comment #2: I appreciate you for your constructive comment. A third column was added to the Table 1 to address to the selection of the algorithm’s parameters. Because the proposed solution are machine learning algorithms, they find the parameters, adaptively. However, there are some values that were specified in this table.

Comment #3:  Finally, I identified that Figure 2 does not have any label. Could you include them?

Respond to Comment #2: I appreciate you for your comment and your attention. The labels for the axis of this figure were added.
